# Performance of glomerular filtration rate equations using serum creatinine in children with congenital anomalies of the kidney and urinary tract

Ferdy Royland Marpaung[1], Santi Wulan Purnami[2], Shofi Andari[2], Ali Rohman[3], Hari Basuki Notobroto[4], Risky Vitria Prasetyo[5], Anggia Augustasia Lumban Toruan[6], Djoko Santoso[7], Aryati Aryati[8]*

**1** Doctoral Program of Medical Science, Faculty of Medicine, Universitas Airlangga, Surabaya, Indonesia, **2** Department of Statistics, Institut Teknologi Sepuluh Nopember, Surabaya, Indonesia, **3** Department of Chemistry, Faculty of Science and Technology, Universitas Airlangga, Surabaya, Indonesia, **4** Faculty of Public Health, Universitas Airlangga, Surabaya Indonesia, **5** Department of Child Health, Faculty of Medicine, Universitas Airlangga, Surabaya, Indonesia, **6** Gotong Royong Hospital, Surabaya, Indonesia, **7** Department of Internal Medicine, Faculty of Medicine, Universitas Airlangga, Surabaya, Indonesia, **8** Department of Clinical Pathology, Faculty of Medicine, Universitas Airlangga, Surabaya, Indonesia

* aryati@fk.unair.ac.id

## Abstract

### Introduction

Congenital anomalies of the kidney and urinary tract (CAKUT) are a significant cause of pediatric morbidity and mortality, often leading to chronic kidney disease (CKD). Accurate glomerular filtration rate (GFR) assessment is crucial for effective management, but a gold-standard pediatric GFR estimation formula remains elusive. This study compared the bedside Schwartz, Chronic Kidney Disease in Children (CKiD-U25) creatinine, and European Kidney Function Consortium (EKFC) creatinine-based equations against measured GFR (mGFR) using technetium-99m-diethylenetriaminepentaacetic acid (Tc-99mDTPA) clearance.

### Methods

Data were collected from 276 children with CAKUT at Dr. Soetomo Academic General Hospital. Estimated GFR (eGFR) was calculated using the bedside Schwartz, CKiD-U25, and EKFC equations. mGFR was determined using Tc-99mDTPA clearance, considered the gold standard. Correlation (Spearman rs), bias, and accuracy (P30, percentage of eGFR within 30% of mGFR) were assessed. Subgroup analysis was performed for children older than two years.

**Data availability statement:** Data are available at supporting information.

**Funding:** AA was supported by Universitas Airlangga, Surabaya, East Java, Indonesia, grant number 345/UN3.LPPM/PT.01.03/2024 under the Penelitian Unggulan Airlangga (PUA) 2024 initiative, and FRM received support from Soetomo Hospital, Surabaya, East Java, Indonesia, with grant number 100.3.3/25.1/102.6/2024 under the Penelitian Unggulan Hospital Scheme.

**Competing interests:** The authors have declared that no competing interests exist.

## Results

Both equations correlated significantly with mGFR (bedside Schwartz: r = 0.793; CKiD-U25: r = 0,793; EKFC: r = 0.745, p < 0.0001). However, the bedside Schwartz systematically underestimated mGFR (bias: −21.7 mL/min/1.73 m²), while the EKFC overestimated (bias: 17 mL/min/1.73 m², p < 0.0001). In children >2 years, correlations strengthened (bedside Schwartz: r = 0.804; EKFC: r = 0.835, p < 0.0001), with the EKFC demonstrating more accuracy (P30: 90.1%) compared to the bedside Schwartz CKiD-U25 creatinine (75% and 75%, respectively)

## Conclusion

These findings suggest the both bedside Schwartz and EKFC could be a reliable tool for GFR estimation in this pediatric CAKUT population.

## Introduction

Congenital anomalies of the kidney and urinary tract (CAKUT) encompass a spectrum of developmental defects affecting the kidneys and urinary tract, representing a significant proportion (approximately 25%) of all congenital anomalies diagnosed in newborns [1]. With an estimated incidence of 3–6 per 1,000 live births, CAKUT are a leading cause of pediatric morbidity and a substantial contributor to renal impairment, including chronic kidney disease (CKD) [2]. Indeed, CKD complicates the clinical course of over half of all children with CAKUT, with approximately 40% ultimately progressing to end-stage renal disease. Consequently, early and accurate assessment of renal function is critical for optimizing patient outcomes.

Glomerular filtration rate (GFR) and urinary albumin excretion are established standard metrics for evaluating kidney function. While measured GFR (mGFR) using exogenous filtration markers, such as technetium-99m-diethylenetriaminepentaacetic acid (99mTc DTPA) clearance, inulin and iohexol, provides the most accurate assessment [3], these techniques are often invasive, resource-intensive, and subject to technical variability [4]. Therefore, estimated GFR (eGFR) based on endogenous markers, particularly creatinine, has become a cornerstone of clinical practice [5]. However, the accuracy of creatinine-based eGFR equations can vary significantly across different populations and demographic subgroups [6]. Recognizing this limitation, Kidney Disease: Improving Global Outcomes (KDIGO) guidelines recommend validation of eGFR equations within specific populations and disease states at minimum of 50 subjects [7]. This study evaluated the performance of two commonly used creatinine-based eGFR equations in children with CAKUT: the widely adopted bedside Schwartz formula (this equation commonly used in our hospital), Chronic Kidney Disease in Children (CKiD-U25) creatinine, and the more recently developed European Kidney Function Consortium (EKFC) formula.

## Materials and methods

### Population study

This retrospective cohort study analyzed electronic medical record (EMR) data from Dr.Soetomo Academic General Hospital to identify pediatric patients with a primary clinical diagnosis of CAKUT between January 1, 2017 and March 15, 2024 (date of data extraction). CAKUT was defined as congenital structural anomalies of kidneys and/or urinary tract documented in the EMR; CAKUT subtypes were abstracted from the treating clinicans' recorded diagnosis and grouped as obstructive and non-obstructive phenotypes (Table 1). Eligible participants were children aged 6 months to 18 years who underwent technetium-99m-diethylenetriaminepentaacetic acid (99mTc DTPA) clearance for mGFR and had serum creatinine assessed on the same day. We excluded records with missing data required to compute eGFR (age, sex, height, serum creatinine, mGFR, and urine albumin-to-creatinine ratio (ACR)). Of 341 CAKUT records screened, 65 (19%) were excluded because a complete paired dataset was not available, most commonly because 99mTc DTPA was not performed, and/or because other key variables (e.g., height or same day creatinine) were missing.

### Measured GFR

At the Department of Nuclear Medicine, Dr.Soetomo Academic General Hospital, Surabaya, East Java, Indonesia, the renal function was assessed using a nuclear renography protocol. Technetium-99m-diethylenetriaminepentaacetic acid (99mTc DTPA) was administered as an intravenous bolus injection [8]. Following established guidelines [9], the 99mTc DTPA dose (mCi) was adjusted based on patient body weight, with subjects positioned supine during administration. Dynamic renal scintigraphy (renography) was then performed using a SPECT/CT scanner (Siemens Symbia-T2 TruePoint) with a low-energy collimator, a 64x64 matrix, and a 20% energy window. Data acquisition proceeded for 30 minutes. The measured GFR (mGFR) using 99mTc DTPA was calculated as the sum of the right and left kidney mGFR values. All mGFR values were normalized to a body surface area of 1.73 m² and expressed as mL/min/1.73 m².

The measuring GFR using Tc99mDTPA is a regularly procedure in the Dr Soetomo Academic General Hospital in evaluating patients with CAKUT. It is often advised when an accurate evaluation of renal function is required to inform clinical decisions, such as prior to an intervention, in the presence of substantial structural abnormalities, or when there is a suspicion of impaired renal function. Consequently, all CAKUT patients underwent Tc-99m DTPA mGFR throughout the cohort duration. Only children with serum creatinine levels assessed on the same day and immediately preceding the nuclear renography procedure were included, ensuring temporal congruence between creatinine-based estimated GFR (eGFR) and measured GFR (mGFR).

### Estimated GFR

The creatinine-based formula, including bedside Schwartz formula and EKFC was compared to mGFR 99mTc DTPA. Blood creatinine concentration was examined using Alinity (Abbott, USA) with enzymatic methods (mg/dL), as described in the previous study [10]. Creatinine testing for this enzymatic method has been calibrated to IDMS (08P65/6K30). The evaluation of each equation was conducted based on the overall number of patients in our study. The necessary data for calculating eGFR Bedside Schwartz, CKiD-U25 formulas and EKFC was obtained from individuals' medical records, which included age, height, sex, and serum creatinine levels. The Bedside Schwartz formula involves multiplying factor 4.13 by height (cm) and dividing the result by blood creatinine concentrations in mg/dL, resulting in data expressed in ml/mnt/1.73 m² [11]. The CKiD-U25 formula involve k factor multiplying height (m) and dividing the blood creatinine. The k factor difference between sex and age, as describe by Pierce et al [12]. For the EKFC, the formula was used without rescaling creatinine (Q), with results in units of mL/min/1.73 m². The equations are as follows: $107.3 \times (SCr/Q)-0.322$ if $SCr < Q$, $107.3 \times (SCr/Q)-1.132$ if $Cr \geq Q$ (ml/min/1.73 m²) [13]. Because EKFC formulas were intended for use for ages > 2 years, we also compared data aged > 2 years between mGFR 99Tcm DTPA and Bedside Schwartz formula, and mGFR 99Tcm DTPA vs EKFC.

**Table 1. Characteristics of the Subjects.**

| Sex | |
|---|---|
| Male (n,%) | 248 (89%) |
| Female (n, %) | 28 (11%) |
| Age (median, IQR) years | 7 (2–11) |
| Height (median, IQR) m | 1.12 (0.85-1.33) |
| Creatinine (median, IQR) (mg/dL) | 0.53 (0.40−0.72) |
| mGFR $^{99}$Tcm DTPA (median, IQR) ml/min/1.73 m$^2$ | 105 (93-122) |
| mGFR $^{99}$Tcm DTPA classification n(%) | |
| <60 ml/min/1.73 | 24 (8.7%) |
| ≥60 ml/min/1.73 | 252 (91.3%) |
| mGFR $^{99}$Tcm DTPA KDIGO classification | |
| G1 ≥90 (n) | 229 |
| G2 60–89 (n) | 23 |
| G3a 45–59 (n) | 6 |
| G3b 30–44 (n) | 3 |
| G4  15–29 (n) | 5 |
| G5 <15 (n) | 10 |
| eGFR median (IQR) ml/min/1.73 m$^2$ | |
| Bedside Schwartz | 83 (68-103) |
| EKFC | 129 (117-143) |
| CKiD-U25 | 84 (80-86) |
| Bedside Schwartz>2 years | 80 (66-98) |
| EKFC>2 years | 126 (116-138) |
| CKiD-U25>2 years | 83 (80-85) |
| Urine Albumin creatinine ratio (n,%) | |
| <30 mg/g | 197 (71) |
| 30-<300 mg/g | 44 (16) |
| ≥300 mg/g | 38 (13) |
| UTI recurrent (n, %) | |
| Yes | 4 (2) |
| No | 272 (98) |
| CAKUT Sub-type (n,%) | |
| Obstructive | 77 (29%) |
| Ureteropelvic junction (UPJ) | 48 (17.9%) |
| Vesicoureteral junction (VUJ) | 26 (9.7%) |
| Posterior urethral valve (PUV) | 3 (1.4%) |
| Non obstructive | 189 (71%) |
| Renal agenesis | 49 (18.4%) |
| Renal cyst | 41 (15.4%) |
| vesicoureteral reflux (VUR) | 33 (12.4%) |
| Double collecting system | 46 (17.1%) |
| Other malformations | 21 (7.7%) |

## Ethical clearance

This study was approved by the Institutional Review Board of Dr. Soetomo Hospital, Surabaya, East Java, Indonesia (ethical clearance number 0398/KEPK/III/2024). Due to the retrospective nature of the study, the requirement for individual patient consent was waived by the Ethics Committee. Data was accessed on March 15, 2024. Data were anonymized, and the authors had no access to identifying patient information at any point during or after data collection.

## Statistical analysis

Patient demographics and baseline characteristics were summarized using descriptive statistics, including frequencies and summary measures. Data were carried out normality tests using the Kolmogorov–Smirnov test with a significant level of $p < 0.05$. The performance of the bedside Schwartz, CKiD-U25, and EKFC equations in estimating GFR was evaluated against mGFR obtained via 99mTc-DTPA clearance using several statistical methods. Spearman's correlation coefficient (r) was calculated to assess the linear relationship between estimated and measured GFR.

Passing-Bablok regression analysis was used to determine the slope and intercept, while bias was assessed using Bland-Altman analysis. Accuracy was evaluated using P30 defined as the percentage of estimated GFR values falling within 30% of the corresponding mGFR 99mTc-DTPA. Statistical analyses were performed using MedCalc® software for Windows (version 20.218). The images of Passing-Bablok regression and Bland-Altman plots and analyses were generated using the Interactive Analytical Comparison and Bias web application (https://bahar.shinyapps.io/method_compare/) [14]. Data between mGFR and eGFR classification were compared as G1, G2, G3a, G3b, G4 and G5, following KDIGO guideline. G1 if GFR > 90 mL/min/1.73 m², G2 if GFR 60–89 mL/min/1.73 m², G3a if GFR 45–59 mL/min/1.73 m², G3b if GFR 30–44 mL/min/1.73 m², G4 if GFR 15–29 mL/min/1.73 m², and G5 if GFR < 15 mL/min/1.73 m². CKD was defined if GFR < 60 mL/min/1.73 m².

# Results

## Baseline characteristics

The study population comprised 276 patients, predominantly male (89%). The median age was 7 years, and the median height was 1.12 meters. The median serum creatinine concentration was 0.53 mg/dL. Thirteen percent of patients exhibited an ACR ≥ 300 mg/g. The median mGFR 99mTc-DTPA was significantly different between the three groups ($p < 0.05$), being lower than the median EKFC-derived eGFR but higher than the median CKiD-U25, and bedside Schwartz-derived eGFR. Four patients (2%) had a history of recurrent urinary tract infections (Table 1).

## Evaluation of eGFR Equations

The bedside Schwartz and CKiD-U25 equations significantly underestimated mGFR 99mTc-DTPA, while the EKFC equation significantly overestimated mGFR 99mTc-DTPA (bias: −21.7 mL/min/1.73 m² for EKFC; 17 mL/min/1.73 m² for bedside Schwartz; 19.9 mL/min/1.73 m² for CKiD-U25; $p < 0.0001$) (Table 2). Despite these biases, both equations demonstrated strong correlations with mGFR 99mTc-DTPA (bedside Schwartz: r = 0.793, $p < 0.0001$; CKiD-U25: r = 0,793 $p < 0.0001$; EKFC: r = 0.745, $p < 0.0001$). In the subgroup of patients older than 2 years, these correlations were even stronger (bedside Schwartz: r = 0.804, $p < 0.0001$; CKiD-U25: r = 0.804 $p < 0.0001$; EKFC: r = 0.835, $p < 0.0001$) (Table 2). Passing–Bablok regression and Bland–Altman plots comparing measured GFR using Tc-99m DTPA with eGFR EKFC, the bedside Schwartz and the CKiD-U25 equations in the full cohort and age > 2 years presented in Fig 1, Fig 2, Fig 3, Fig 4, Fig 5, and Fig 6.

Concordance with mGFR 99mTc-DTPA was higher for the EKFC equation compared to the bedside Schwartz equation (83.2% vs. 68.5%, $p < 0.0001$). In the subgroup older than 2 years, two equation demonstrated difference accuracy (EKFC, P30: 90.1%, the bedside Schwartz equation, P30 75%). Furthermore, the EKFC equation yielded lower root mean square error (RMSE) and median difference values compared to the bedside Schwartz equation (Table 2).

**Table 2.  Comparison mGFR ⁹⁹Tcm DTPA to bedside Schwartz and EKFC.**

| Metric | Schwartz | CKiD-U25 | EKFC |
|---|---|---|---|
| Bias (ml/min/1.73m²) | −19.92 | −19.71 | 20.75 |
| RMSE | 31.00 | 30.90 | 26.34 |
| P30 | 0.703 | 0.707 | 0.772 |
| Lin's CCC | 0.686 | 0.688 | 0.784 |
| Slope | 0.891 | 0.894 | 1.072 |
| Intercept | −8.39 | −8.41 | 13.12 |
| Correlation (r) | 0.793 | 0.793 | 0.916 |
| R² | 0.181 | 0.187 | 0.409 |

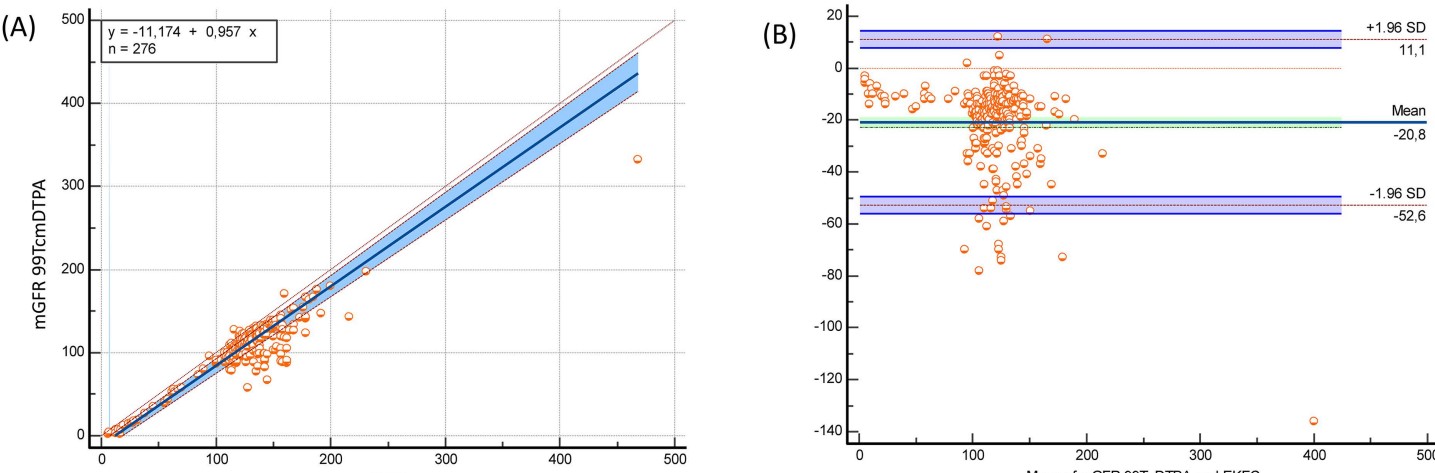

**Fig 1.  Passing–Bablok regression and Bland–Altman analysis comparing EKFC eGFR with measured GFR (Tc-99m DTPA) in the full cohort (n = 276). (A)** The Passing–Bablok regression shows the relationship between EKFC-estimated GFR and mGFR99mTc-DTPA. The regression equation (y = −11.174 + 0.957x) demonstrates a slope slightly below unity, indicating mild proportional underestimation by the EKFC equation at higher GFR values. The 95% confidence interval band (shaded blue) reflects narrow dispersion, particularly between GFR 50–200 mL/min/1.73 m², where most data points cluster. The identity line (x = y) is shown in red for reference. **(B)** The Bland–Altman plot displays the bias (mean difference: −20.8 mL/min/1.73 m²) and limits of agreement (−52.6 to +11.1 mL/min/1.73 m²). Most points fall within limits, but a consistent negative bias indicates systematic underestimation of GFR by EKFC relative to mGFR. One high-GFR outlier (>400 mL/min/1.73 m²) is visible.

Discrepancies were observed in the classification of patients into G1 and G2 categories when comparing EKFC-derived eGFR versus mGFR 99mTc-DTPA and bedside Schwartz-derived eGFR versus mGFR 99mTc-DTPA. The bedside Schwartz equation classified five times more patients as G2 compared to the EKFC equation. Categorical agreement was higher for the EKFC equation. However, no difference was observed between the two equations in the number of patients classified as having CKD (Table 3).

## Discussion

It is known that CAKUT lowers the number of nephrons at birth, scars the kidneys, and leads to end-stage renal disease [15–17]. Advanced CKD is associated with a high risk of mortality and morbidity. Previous research found that 4.5% of children with CKD advance to end-stage renal disease (ESRD) in the first year, 10.3% in the second year, and 23.9% in the fifth [15–17]. Children with ESRD have a 30 times greater chance of death and are

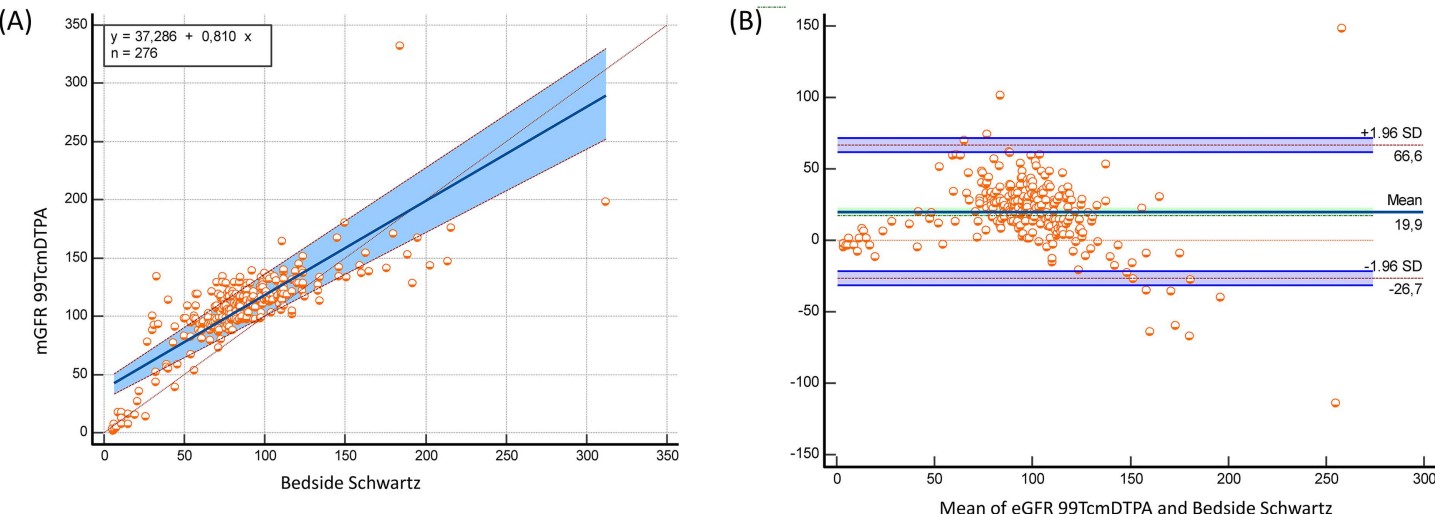

**Fig 2. Passing–Bablok regression and Bland-Altman analysis comparing bedside Schwartz eGFR with measured GFR (Tc-99m DTPA) in the full cohort (n = 276). (A)** The Passing–Bablok regression (y = 37.286 + 0.810x) indicates substantial constant bias (intercept ≈ 37 mL/min/1.73 m²) and a slope <1.0, showing that the bedside Schwartz equation progressively underestimates mGFR at higher values. The wide blue-shaded confidence band highlights increased variability at higher GFRs. **(B)** The Bland–Altman plot shows mean positive bias (+19.9 mL/min/1.73 m²), indicating that bedside Schwartz overestimates GFR overall in this population. Limits of agreement range from –26.7 to +66.6 mL/min/1.73 m². Several scattered residuals at low GFR values indicate broad dispersion and lower precision.

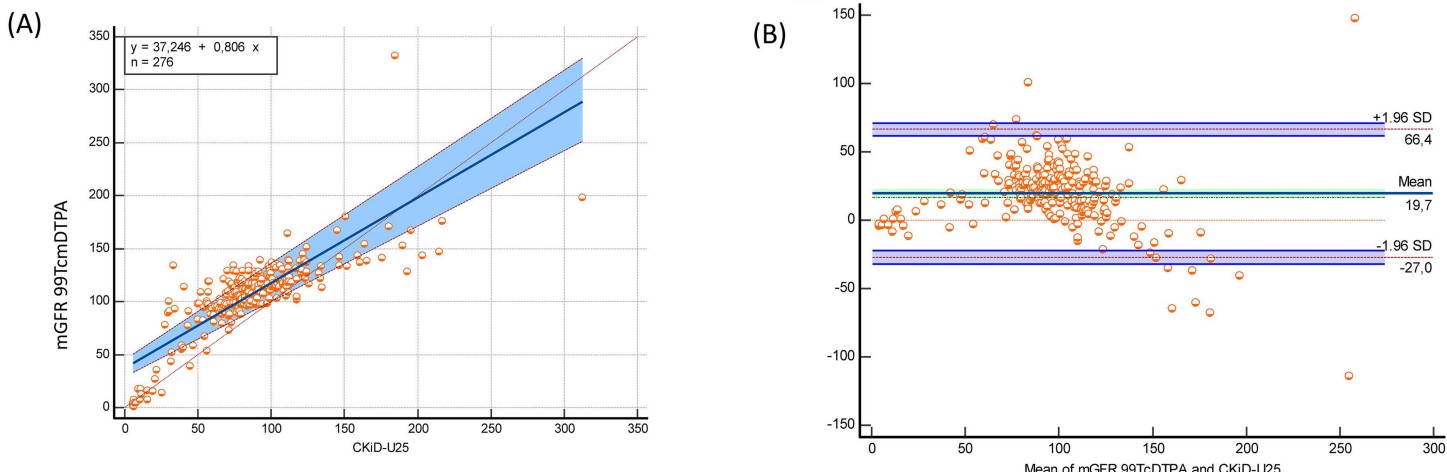

**Fig 3. Passing–Bablok regression and Bland–Altman analysis comparing CKiD-U25 creatinine eGFR with measured GFR (Tc-99m DTPA) in the full cohort (n = 276) (A)** Passing-bablok regression analysis (y = 37.246 + 0.806x) shows constant and proportional bias. The slope <1.0 suggests GFR underestimation at higher ranges. The confidence band widens moderately toward high GFR values (>200 mL/min/1.73 m²), reflecting greater uncertainty. **(B)** Bland–Altman analysis indicates mean bias of +19.7 mL/min/1.73 m², with LOA between –27.0 and +66.4 mL/min/1.73 m².

more likely to require long-term renal replacement therapy, which has a psychosocial, emotional, and economic burden on these patients and their families [18]. This is crucial for establishing an early diagnosis of kidney disorders, particularly in children with CAKUT, and for determining the appropriate dosage of treatment that takes into account kidney function. Given the importance of using eGFR which is very simple in determining kidney filtration

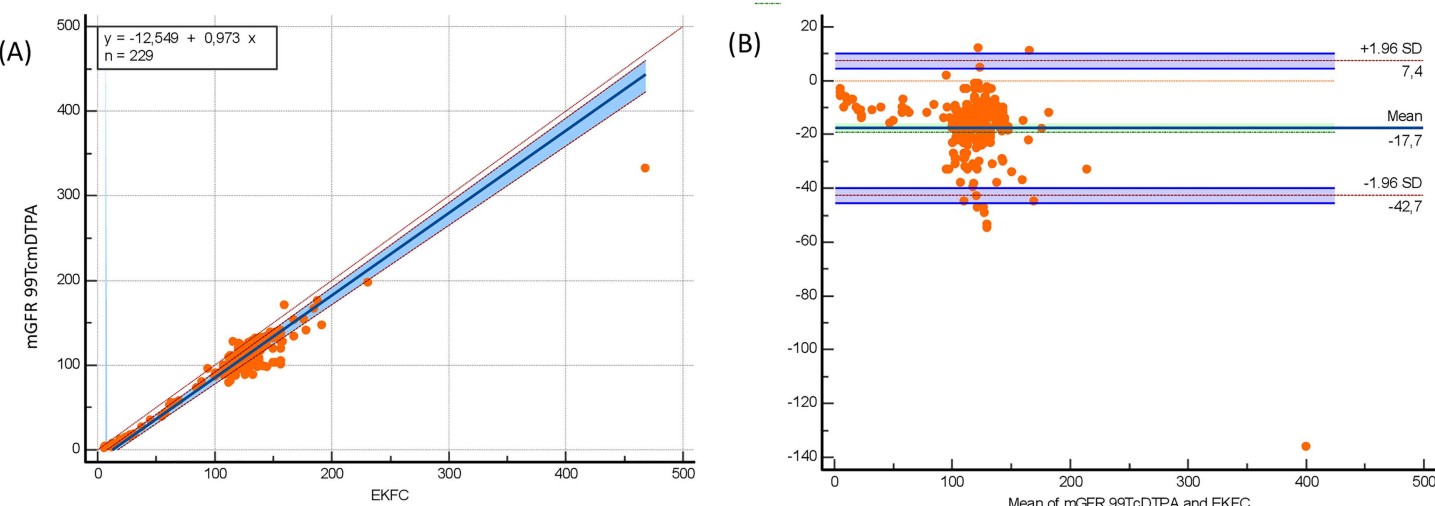

**Fig 4. Passing–Bablok regression and Bland-Altman analysis comparing EKFC eGFR with measured GFR (Tc-99m DTPA) in children aged >2 years (n=229). (A)** Passing-Bablok regression (y=−12.549+0.973x) demonstrates strong agreement, with slope near 1.0 and narrow confidence intervals. Compared with the full cohort, dispersion around the identity line is reduced, suggesting improved performance of EKFC in children >2 years. **(B)** Bland–Altman analysis reveals mean bias of −17.7 mL/min/1.73 m² with LOA from −42.7 to +7.4 mL/min/1.73 m². Variability is notably lower than in the full cohort, indicating the EKFC equation is more accurate and precise when applied to children older than 2 years.

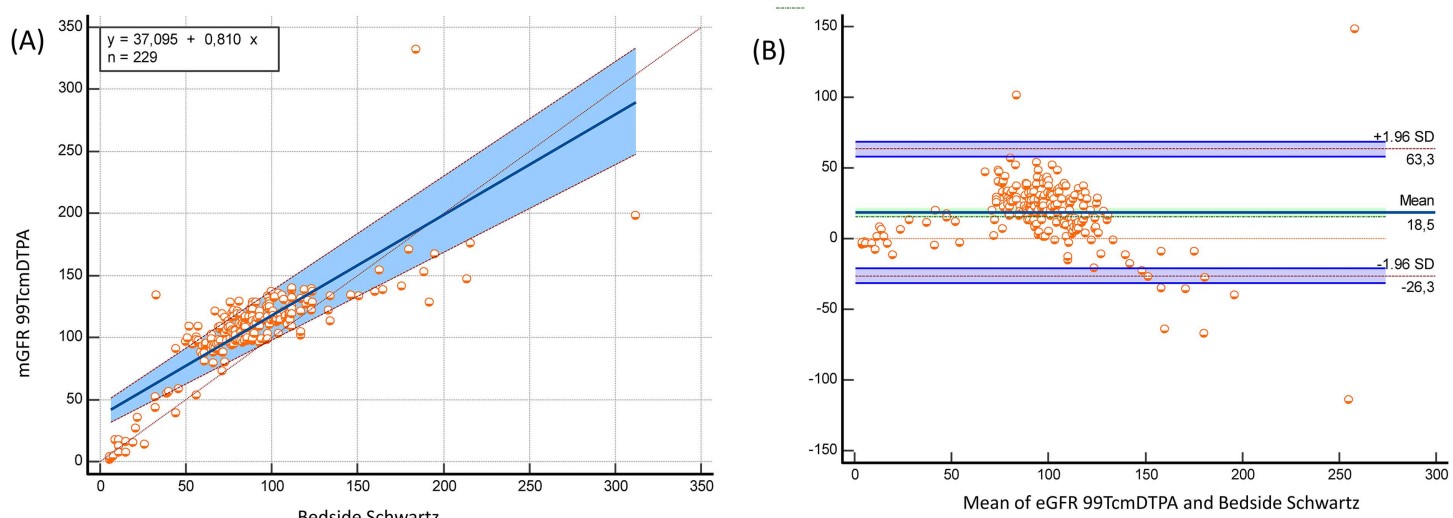

**Fig 5. Passing–Bablok regression and Bland–Altman analysis comparing bedside Schwartz eGFR with measured GFR (Tc-99m DTPA) in children aged >2 years (n=229). (A)** The Passing-Bablok regression equation (y=37.095+0.810x) maintains a similar bias structure as in the full cohort, but with reduced scatter. The shaded confidence band remains wider than EKFC, illustrating lower precision in this age group. **(B)** Bland–Altman results show mean bias of +18.5 mL/min/1.73 m² and LOA from −26.3 to +63.3 mL/min/1.73 m². The persistent positive bias indicates overestimation of GFR by bedside Schwartz, particularly between 80–150 mL/min/1.73 m² where most data points lie.

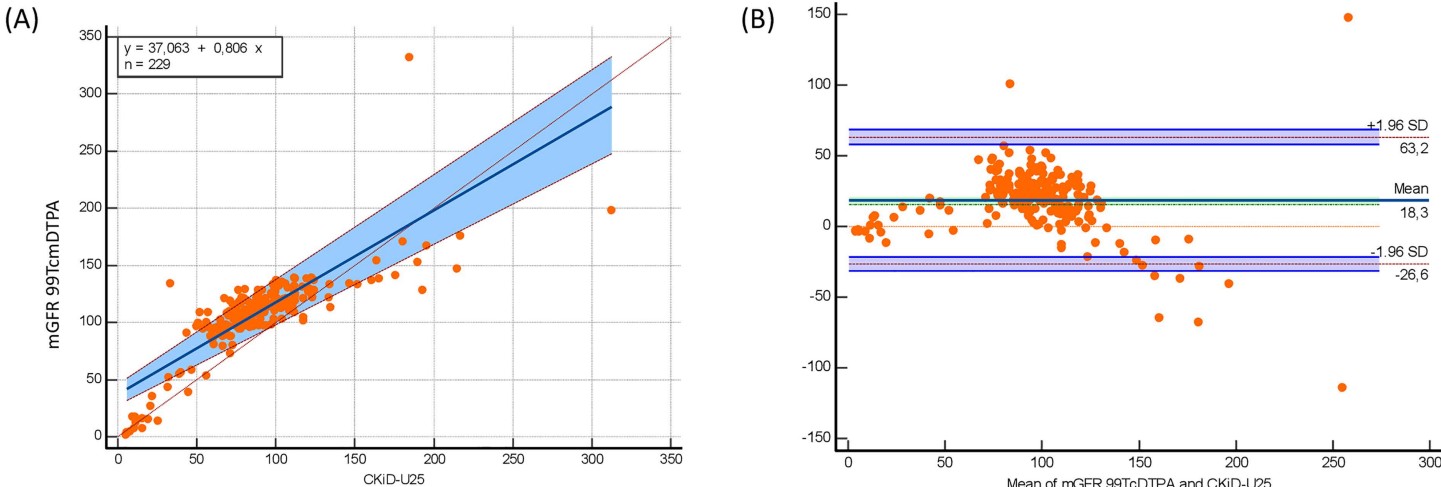

**Fig 6. Passing–Bablok regression and Bland–Altman analysis comparing CKiD-U25 eGFR with measured GFR (Tc-99m DTPA) in children aged >2 years (n = 229). (A)** Passing-Bablok regression (y = 37.063 + 0.806x) shows nearly identical behavior to bedside Schwartz, with similar intercept and slope. Prediction intervals widen at high GFR values, indicating less stability beyond 150–200 mL/min/1.73 m². **(B)** Bland–Altman analysis indicates mean bias of +18.3 mL/min/1.73 m² and LOA from −26.6 to +63.2 mL/min/1.73 m².

**Table 3. Classification of kidney function comparison mGFR to EKFC, bedside Schwartz and CKiD-U25.**

|  | mGFR Category | G1 (≥90) | G2 (60–89) | G3a (45–59) | G3b (30–44) | G4 (15–29) | G5 (<15) | Total | Agreement (%) |
|---|---|---|---|---|---|---|---|---|---|
| **EKFC** | G1 (≥90) | 229 | 0 | 0 | 0 | 0 | 0 | 229 | 86.2 |
|  | G2 (60–89) | 21 | 2 | 0 | 0 | 0 | 0 | 23 |  |
|  | G3a (45–59) | 1 | 5 | 0 | 0 | 0 | 0 | 6 |  |
|  | G3b (30–44) | 0 | 0 | 3 | 0 | 0 | 0 | 3 |  |
|  | G4 (15–29) | 0 | 0 | 0 | 3 | 2 | 0 | 5 |  |
|  | G5 (<15) | 0 | 0 | 0 | 0 | 5 | 5 | 10 |  |
|  | Total | 251 | 7 | 3 | 3 | 7 | 5 | 276 |  |
| **Bedside Schwartz** | G1 (≥90) | 106 | 106 | 11 | 6 | 0 | 0 | 229 | 49.6 |
|  | G2 (60–89) | 0 | 16 | 4 | 2 | 1 | 0 | 23 |  |
|  | G3a (45–59) | 0 | 0 | 2 | 4 | 0 | 0 | 6 |  |
|  | G3b (30–44) | 0 | 0 | 0 | 2 | 1 | 0 | 3 |  |
|  | G4 (15–29) | 0 | 0 | 0 | 0 | 3 | 2 | 5 |  |
|  | G5 (<15) | 0 | 0 | 0 | 0 | 2 | 8 | 10 |  |
|  | Total | 106 | 122 | 17 | 14 | 7 | 10 | 276 |  |
| **CKiD-U25** | G1 (≥90) | 106 | 106 | 11 | 6 | 0 | 0 | 229 | 49.6 |
|  | G2 (60–89) | 0 | 16 | 4 | 1 | 2 | 0 | 23 |  |
|  | G3a (45–59) | 0 | 0 | 2 | 4 | 0 | 0 | 6 |  |
|  | G3b (30–44) | 0 | 0 | 0 | 2 | 1 | 0 | 3 |  |
|  | G4 (15–29) | 0 | 0 | 0 | 0 | 3 | 2 | 5 |  |
|  | G5 (<15) | 0 | 0 | 0 | 0 | 2 | 8 | 10 |  |
|  | Total | 106 | 122 | 17 | 13 | 8 | 10 | 276 |  |

function, it is necessary to analyze routinely used formulas, namely Schwartz, CKID-U25 and the recently equation, namely EKFC.

While established risk factors for CKD progression in pediatric populations are well-documented a universally accepted and standardized GFR estimation equation specifically for children with CAKUT remains an unmet need. A key distinction between the EKFC, CKID-U25, and bedside Schwartz equations lies in their respective independent variables [12]. The bedside Schwartz and CKID-U25 equations incorporates height, whereas the EKFC equation utilizes age. From a practical standpoint, the use of age simplifies eGFR reporting in clinical laboratories, as it eliminates the requirement for height measurements, thereby streamlining workflow and potentially reducing measurement error.

This study assessed the performance three equations; the bedside Schwartz and EKFC equations for estimating GFR in a cohort of children with CAKUT. Our findings demonstrate that both equations exhibit strong correlations with mGFR as measured by 99mTc-DTPA clearance, aligning with previous findings. Zhao et al. [19] reported similar strong correlations and good agreement between the EKFC equation and mGFR in a Chinese cohort of patients with chronic kidney disease. Likewise, Bhowmick et al. observed an intraclass correlation of 74% and an accuracy (eGFR within 30% of mGFR) of 78.8% in critically ill children [20]. However, this study represents, to our knowledge, the first evaluation of the EKFC equation's performance specifically within a CAKUT population.

In the current study, the accuracy of p30 for all equations was more than 75%. This percentage showed that both formulas can be used clinically [7,21]. However, p30 is not yet ideal. A good formula can include p20 or p10, which can cover not only 75% but ideally 100% of the comparison with the mGFR. In this study we found p30 only cover 75% of accuracy. It mean the target for p20 and p10 should be lower statistically. Nevertheless, targets for p20 and p10 are challenging. The fact that the RMSE and median difference were the same for both eGFRs showed that they both may be used, with more accurate data from EKFC.

The prevalence of obstructive CAKUT was higher than obstructive CAKUT in our cohort. This finding in-line with Puspitarini study on CAKUT prevalence in Jakarta region [22]. In addition, this study a striking sex-based disparity in the prevalence of CAKUT, with males exhibiting a significantly higher incidence (89%) compared to females. The median age at evaluation in our tertiary referral hospital was 7 years, reflecting the lower access of prenatal imaging studies in Indonesia. This finding contrasts with a previous study that reported a lower male prevalence of 60% [23]. However, our observation aligns with a similar study conducted in Nigeria, which also demonstrated a higher male preponderance (70.9%) [24]. Despite these consistent findings, the underlying etiological mechanisms driving this sex-based disparity in CAKUT remain elusive and warrant further investigation.

Almost nine percent of the cohort had an mGFR < 60 mL/min/1.73 m² at the time of evaluation, corresponding to at least CKD stage 3 by GFR criteria, significantly lower compared to other studies that showed a range from 12% to 21.3% [24]. This difference suggests that the CAKUT cohort in the study may have lower risk of developing CKD compared to other CAKUT populations. Furthermore, the difference in the eGFR equation, method and sample size used between those study may affected to this varied results.

We performed a sensitivity analysis to assess the impact of extreme discordant cases on performance metrics by excluding children whose KDIGO GFR categories varied by three or more stages between measured and estimated GFR. No such outliers were identified for the EKFC equation, whereas seven outliers were identified for the bedside Schwartz and CKID-U25 equations. The clinical profiles of these outliers shared features that are known to challenge creatinine-based GFR estimation, including: very low or very high muscle mass relative to body size (e.g., children with severe growth delay or chronic illness), extremely abnormal serum creatinine values out of proportion to stable structural renal damage, suggesting either non–steady-state kidney function or possible intercurrent illness around the time of measurement, and early infancy with rapidly changing GFR and creatinine generation, where the maturation of renal function and creatinine production is highly dynamic.

Recalculation of accuracy metrics after removing these cases showed minimal change for EKFC (bias 20.81 vs. 20.75 mL/min/1.73 m²; RMSE 26.48 vs. 26.34; CCC 0.786 vs. 0.784; P30 0.770 vs. 0.772), confirming that EKFC performance was robust across the entire cohort. Further evaluation for bedside Schwartz and CKiD-U25 showed noted improvement when isolated data were excluded (RMSE reduced from 31.00 to 29.42; $R^2$ was risen from 0.18 to 0.28; CCC increased from 0.686 to 0.711). Those data indicate that the bedside Schwartz equation is more likely sensitive to extreme discordance, whereas the EKFC equation maintains stable performance across the full range of CAKUT phenotypes.

## Limitations

This study possesses several limitations. The retrospective design inherently restricts the ability to establish causality and introduces the potential for selection bias. Furthermore, the absence of a population-specific Q creatinine value within the EKFC formula constitutes a limitation. Future research should investigate the impact of rescaling Q creatinine using median creatinine values derived from a healthy pediatric population. As demonstrated by Nyman et al. [25], appropriate Q creatinine values are crucial for optimizing the performance of GFR estimating equations across diverse populations with varying creatinine generation. Given regional variations in creatinine levels observed in Indonesian adults [26], further investigation of pediatric creatinine levels across different regions of Indonesia is warranted.

Approximately 19% of children with CAKUT did not undergo Tc-99m DTPA, primarily due to parental refusal of nuclear medicine procedures, logistic constraints, or scheduling issues rather than suspicion of discrepant eGFR. Therefore, some degree of selection bias cannot be excluded. Those who accepted a nuclear medicine procedure may differ in unmeasured ways (e.g., disease severity, family preference, or socioeconomic factors) from those who declined. However, we emphasized that the clinical indication for Tc-99m DTPA was driven by the presence of CAKUT and need for accurate GFR, rather than by prior suspicion of a discrepancy between eGFR and mGFR. Therefore, while selection bias may exist, we believe it is unlikely to be systematically biased toward highly discordant eGFR/mGFR pairs.

While Tc-99m DTPA clearance was utilized as a measure of mGFR [27], iohexol clearance is increasingly recognized as a preferred reference method with guideline from the EKFC recommending its use for assessing renal function due to its demonstrated cost-effectiveness and safety in both multi- and mono-sampling protocols [28]. Future studies employing iohexol clearance as the mGFR standard are therefore encouraged.

Finally, while creatinine-based eGFR is widely accessible, its accuracy can be compromised in specific populations, such as those with limb amputations or malnutrition [29], conditions potentially relevant to patients with CAKUT and other comorbidities. Therefore, future research exploring the utility of cystatin C-based or combined creatinine-cystatin C equations (9) for eGFR estimation in this population is warranted. While no single "ideal" GFR estimation formula exists, a thorough understanding of the strengths and limitations of each approach is essential for selecting the most appropriate method for precise assessment of kidney function in individual patients.

## Conclusions

Both the bedside Schwartz and EKFC equations demonstrated strong correlations with mGFR derived from Tc-99m DTPA clearance in pediatric CAKUT patients. The EKFC exhibited more accuracy, particularly in children older than two years. These findings suggest that all equations may offer a more refined estimation of GFR in this population. However, given the limitations of creatinine-based estimations and the potential for improved accuracy with alternative mGFR markers, future investigations employing iohexol clearance as the reference standard are warranted to further refine our understanding of optimal GFR assessment in children with CAKUT.

## Supporting information

**S1 Data. Data sharing.**
(XLSX)

## Acknowledgments

The authors express their gratitude to nuclear medicine department for their substantial data support in this study. Also our special thanks to Annisa Feby Canintika, MD for her substantial support in this manuscript.

## Author contributions

**Conceptualization:** Ferdy Royland Marpaung, Santi Wulan Purnami, Ali Rohman, Hari Basuki Notobroto, Risky Vitria Prasetyo, Djoko Santoso, Anggia Augustasia Lumban Toruan, Aryati Aryati.

**Data curation:** Ferdy Royland Marpaung, Risky Vitria Prasetyo.

**Formal analysis:** Ferdy Royland Marpaung, Santi Wulan Purnami, Shofi Andari, Hari Basuki Notobroto, Risky Vitria Prasetyo, Djoko Santoso, Aryati Aryati.

**Funding acquisition:** Ferdy Royland Marpaung, Djoko Santoso, Aryati Aryati.

**Investigation:** Ferdy Royland Marpaung, Anggia Augustasia Lumban Toruan, Aryati Aryati.

**Methodology:** Ferdy Royland Marpaung, Santi Wulan Purnami, Shofi Andari, Ali Rohman, Hari Basuki Notobroto, Risky Vitria Prasetyo, Djoko Santoso, Anggia Augustasia Lumban Toruan, Aryati Aryati.

**Project administration:** Ferdy Royland Marpaung.

**Resources:** Ferdy Royland Marpaung.

**Software:** Ferdy Royland Marpaung.

**Supervision:** Hari Basuki Notobroto, Djoko Santoso, Aryati Aryati.

**Validation:** Ferdy Royland Marpaung, Santi Wulan Purnami, Shofi Andari, Ali Rohman, Risky Vitria Prasetyo, Anggia Augustasia Lumban Toruan, Aryati Aryati.

**Visualization:** Ferdy Royland Marpaung.

**Writing – original draft:** Ferdy Royland Marpaung, Aryati Aryati.

**Writing – review & editing:** Ferdy Royland Marpaung, Santi Wulan Purnami, Shofi Andari, Ali Rohman, Hari Basuki Notobroto, Risky Vitria Prasetyo, Djoko Santoso, Anggia Augustasia Lumban Toruan, Aryati Aryati.

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
