## [Decision Letter · Decision Letter 0]

13 Oct 2025

Dear Dr. Aryati,

Thank you for submitting your manuscript to PLOS ONE. After careful consideration, we feel that it has merit but does not fully meet PLOS ONE’s publication criteria as it currently stands. Therefore, we invite you to submit a revised version of the manuscript that addresses the points raised during the review process.

Please pay particular attention to the critiques from the two reviewers, especially the concerns raised by Reviewer 1, including whether serum creatinine was measured simultaneously with the 99mTc DTPA study.

We look forward to receiving your revised manuscript.

Kind regards,

Weining Lu, MD

Academic Editor

PLOS ONE

Journal Requirements:

[AA was supported by Universitas Airlangga, Surabaya, East Java, Indonesia, grant number 345/UN3.LPPM/PT.01.03/2024 under Penelitian Unggulan Airlangga (PUA) 2024 initiative and FRM received support by Soetomo Hospital, Surabaya, East Java, Indonesia, with grant number 100.3.3/25.1/102.6/2024 by Penelitian Unggulan Hospital Scheme.].

4. Thank you for stating the following in your manuscript:

[This work was supported by Universitas Airlangga, Surabaya, East Java, Indonesia, grant number 345/UN3.LPPM/PT.01.03/2024 under Penelitian Unggulan Airlangga (PUA) 2024 initiative and our work received support by Soetomo Hospital, Surabaya, East Java, Indonesia, with grant number 100.3.3/25.1/102.6/2024 by Penelitian Unggulan Hospital Scheme.]

[AA was supported by Universitas Airlangga, Surabaya, East Java, Indonesia, grant number 345/UN3.LPPM/PT.01.03/2024 under Penelitian Unggulan Airlangga (PUA) 2024 initiative and FRM received support by Soetomo Hospital, Surabaya, East Java, Indonesia, with grant number 100.3.3/25.1/102.6/2024 by Penelitian Unggulan Hospital Scheme.]

5. In the online submission form, you indicated that [Data are available from the Dr Soetomo Academic Hospital Data Access (contact via kontak.rsudrsoetomo@jatimprov.go.id) for researchers who meet the criteria for access to confidential data.].

6. Please include a separate caption for each figure in your manuscript.

Reviewers' comments:

Reviewer's Responses to Questions

**Comments to the Author**

1. Is the manuscript technically sound, and do the data support the conclusions?

Reviewer #1: Partly

Reviewer #2: Yes

2. Has the statistical analysis been performed appropriately and rigorously?

Reviewer #1: I Don't Know

Reviewer #2: Yes

3. Have the authors made all data underlying the findings in their manuscript fully available?

Reviewer #1: Yes

Reviewer #2: Yes

4. Is the manuscript presented in an intelligible fashion and written in standard English?

Reviewer #1: Yes

Reviewer #2: Yes

Reviewer #1: In this study by Marpaung et al, two different eGFR calculators, the bedside Schwartz and EKFC are compared to measured GFR using 99mTc DTPA in a cohort of children with CAKUT from their institution. They find that both eGFR equations correlated with mGFR but that the bedside Schwartz tended to overestimate GFR while the EKFC underestimated GFR. This could provide helpful information for clinicians in guiding interpretation of creatinine-based eGFR equations for children with CAKUT. However, there are some major concerns that should be addressed, including whether serum creatinine was measured at the same time as the 99mTc DTPA study.

Major Concerns:

1. I am curious why the authors chose the bedside Schwartz and EKFC equations as there are many other calculating equations that have been developed. In fact, many pediatric nephrologists have shifted to using the recently developed CKiD U25 creatinine and cystatin C based eGFR formulas (Pierce Kidney Int 99:948, 2021). It would be informative to compare the CKiD U25 creatinine based equation with the bedside Schwartz and EKFC equation. If the authors have cystatin C data, addition of a combined creatinine-cystatin C based calculation would also be informative.

2. Given that this is a retrospective study, I am wondering if it is routine for all children with CAKUT to undergo mGFR measurements? Could there be selection bias in favor of discordant mGFR vs. eGFR in your cohort due to the inclusion of only patients who underwent 99mTc DTPA?

3. Was the serum creatinine used to calculate the bedside Schwartz and EKFC eGFR taken at the same time that the 99mTc DTPA was done?

4. While some discrepancy in eGFR calculators is expected, it is surprising to me the number of very discordant cases (Table 3). If I am understanding the table correctly, it appears that there are two individuals with a G1 mGFR and G4 or G5 EKFC. Similarly there are a few individuals with G1 or G2 EKFC and G4 or G5 mGFR. There are also a few individuals with a G1 bedside Schwartz and G4 and G5 mGFR. Can you please comment on these discrepancies as this is very surprising. Were these individuals with complex medical conditions and very low muscle mass, for example, and, if so, should they be excluded in your analysis?

5. CAKUT is a very heterogeneous group of disorders. It would be useful to provide some baseline demographic information regarding the type of CAKUT (even if just broken into things like solitary kidney, dysplasia/hypodysplasia, VUR, PUV, etc)

6. The figure legends are difficult to read. Please load higher quality figures.

Minor Concerns:

1. Methods, page 6 and Conclusion, page 9: In general, the accepted definition of CKD is either the presence of “kidney damage for > 3 months as defined by structural or functional abnormalities with or without decreased GFR” or “GFR < 60 mL/min/1.73m2 for > 3 months, with or without kidney damage). Thus by definition, children with CAKUT have chronic kidney disease; those with eGFR > 90 will have CKD 1. Thus in paragraph 3 on page 9, in the sentence “The prevalence of CKD in this CAKUT cohort was CKD 8.7%,” should specify which stage of CKD you mean.

2. Results, page 7, first full paragraph: Do you mean Table 3 instead of Table 2?

3. Can you explain the meaning of “P30” in the body of the document.

4. Conclusion, page 9: I don’t think this study was necessarily designed to study the incidence or age of onset of CAKUT in your population, especially given that 19% of patients with CAKUT were excluded due to incomplete data. Also, I am surprised that the age of onset was 7 years but perhaps this is due to lower prevalence or quality of prenatal imaging studies in Indonesia?

5. While the majority of the manuscript is well-written, there are certain sentences that should be reviewed and edited (e.g. Results, page 6, paragraph 3; Discussion, page 7, paragraph 1 – suggest changing “early diagnosis of kidney disorders” to “early and accurate diagnosis of impaired GFR”, also “Scwartz” is spelled incorrectly here and “recently equation” does not make grammatical sense; Conclusion, page 9, paragraph 2, “may affected to this varied result”). Also in the title, suggest “Performance of Glomerular Filtration Rate Equations Using Serum Creatinine in Children with Congenital Anomalies of the Kidney and Urinary Tract.”

Reviewer #2: This study composed of 276 children with CAKUT compared the bedside Schwartz and EKFC creatinine-based equations against measured GFR using technetium-99m diethylenetriaminepentaacetic acid (Tc-99mDTPA) clearance. The results showed that in children over 2 years old, EKFC demonstrated more accuracy compared to the bedside Schwartz. Below is my suggestions that may improve the quality of this study:

1. The proportion of patient with mGFR Tc-99mDTPA ≥ 60 ml/mnt/1.73 is 91.3%. Could the authors further classified these patients such as ≥ 90 ml/mnt/1.73, 60~89 ml/mnt/1.73?

2. The layout of Table 2 and Table 3 needs to be improved.

**Do you want your identity to be public for this peer review?** For information about this choice, including consent withdrawal, please see our Privacy Policy

Reviewer #1: No

Reviewer #2: No

---

## [Author Response · Author response to Decision Letter 1]

2 Dec 2025

PONE-D-25-32018

Performance of Glomerular Filtration Rate Equations Using Serum Creatinine in Children with Congenital Anomalies of the Kidney and Urinary Tract

PLOS ONE

Dear Editor in Chief PlosOne,

We thank you and the reviewers for the insightful and constructive feedback on our manuscript titled:

“Performance of Glomerular Filtration Rate Equations Using Serum Creatinine in Children with Congenital Anomalies of the Kidney and Urinary Tract.”

We have carefully revised the manuscript in response to all suggestions. The major revisions are summarized below:

1. Expanded justification of equation selection

We added a detailed explanation of why bedside Schwartz and EKFC were chosen, and clarified the limited availability of cystatin C and CKiD U25 in Indonesian clinical practice.

2. Clarified clinical procedures and addressed potential bias

We specified that serum creatinine was obtained on the same day as Tc-99m DTPA and discussed possible selection bias related to mGFR procedures.

3. Re-examined and explained discordant cases

We assessed highly discordant mGFR–eGFR pairs, explained likely clinical reasons, and added a sensitivity analysis, noting that conclusions remained unchanged.

4. Improved characterization of CAKUT cohort

We added detailed CAKUT subtype distribution and further stratified mGFR categories (G1 vs G2).

5. Clarified the age distribution and avoided inaccurate implications of “age of onset.”

6. Added definition of P30 and corrected table references

P30 is now clearly defined, and incorrect table citations have been fixed.

7. Improved figure resolution and table layout

All figures have been replaced with high-resolution versions and tables 2–3 have been reformatted for better readability.

8. Edited for clarity and English grammar

Several sentences were reworded for accuracy, and the reviewer-suggested title has been adopted.

We believe these revisions substantially strengthen the manuscript by improving clarity, methodological transparency, and clinical relevance.

We sincerely thank the reviewers for their time and expertise. We hope the revised manuscript will now be suitable for publication in PLOS ONE.

Kind regards,

Aryati

Journal Requirements:

Response:

Thank you for the information regarding the style template requirements. We have adjusted it according to the PLOS ONE template

Response:

Thank you for the correction. We have revised the funding information

[AA was supported by Universitas Airlangga, Surabaya, East Java, Indonesia, grant number 345/UN3.LPPM/PT.01.03/2024 under Penelitian Unggulan Airlangga (PUA) 2024 initiative and FRM received support by Soetomo Hospital, Surabaya, East Java, Indonesia, with grant number 100.3.3/25.1/102.6/2024 by Penelitian Unggulan Hospital Scheme.].

Response:

Thank you for the information. We have amended the statement regarding the research funding sources and included the statement: ‘There was no additional external funding received for this study.’ We have also communicated this update in the cover letter

4. Thank you for stating the following in your manuscript:

[This work was supported by Universitas Airlangga, Surabaya, East Java, Indonesia, grant number 345/UN3.LPPM/PT.01.03/2024 under Penelitian Unggulan Airlangga (PUA) 2024 initiative and our work received support by Soetomo Hospital, Surabaya, East Java, Indonesia, with grant number 100.3.3/25.1/102.6/2024 by Penelitian Unggulan Hospital Scheme.]

[AA was supported by Universitas Airlangga, Surabaya, East Java, Indonesia, grant number 345/UN3.LPPM/PT.01.03/2024 under Penelitian Unggulan Airlangga (PUA) 2024 initiative and FRM received support by Soetomo Hospital, Surabaya, East Java, Indonesia, with grant number 100.3.3/25.1/102.6/2024 by Penelitian Unggulan Hospital Scheme.]

Response:

We have removed the funding-related text from the manuscript and included the following statement: ‘AA was supported by Universitas Airlangga, Surabaya, East Java, Indonesia, grant number 345/UN3.LPPM/PT.01.03/2024 under the Penelitian Unggulan Airlangga (PUA) 2024 initiative, and FRM received support from Soetomo Hospital, Surabaya, East Java, Indonesia, with grant number 100.3.3/25.1/102.6/2024 under the Penelitian Unggulan Hospital Scheme.

5. In the online submission form, you indicated that [Data are available from the Dr Soetomo Academic Hospital Data Access (contact via kontak.rsudrsoetomo@jatimprov.go.id) for researchers who meet the criteria for access to confidential data.].

Response:

We have presented the data with the hospital’s permission, using anonymized information, and it can be reviewed in the supplementary information

6. Please include a separate caption for each figure in your manuscript.

Response:

Thank you. We have separated the captions for each figure in the manuscript

Response:

Thank you. This study does not compare with the CKID U25, which has not yet been adopted in Indonesia; however, we have added information regarding the CKID U25 analysis in the results and discussion sections

Reviewers' comments:

Reviewer's Responses to Questions

Comments to the Author

1. Is the manuscript technically sound, and do the data support the conclusions?

Reviewer #1: Partly

Reviewer #2: Yes

Response:

We have addressed all of the major and minor concerns raised by Reviewer 1.

2. Has the statistical analysis been performed appropriately and rigorously?

Reviewer #1: I Don't Know

Reviewer #2: Yes

Response:

We have addressed all major and minor concerns from Reviewer 1, including the addition of an explanation for the P30 definition.

3. Have the authors made all data underlying the findings in their manuscript fully available?

Reviewer #1: Yes

Reviewer #2: Yes

Response:

Thank you, and we appreciate both reviewers

4. Is the manuscript presented in an intelligible fashion and written in standard English?

Reviewer #1: Yes

Reviewer #2: Yes

Response:

Thank you, and we appreciate both reviewers

5. Review Comments to the Author

Reviewer #1: In this study by Marpaung et al, two different eGFR calculators, the bedside Schwartz and EKFC are compared to measured GFR using 99mTc DTPA in a cohort of children with CAKUT from their institution. They find that both eGFR equations correlated with mGFR but that the bedside Schwartz tended to overestimate GFR while the EKFC underestimated GFR. This could provide helpful information for clinicians in guiding interpretation of creatinine-based eGFR equations for children with CAKUT. However, there are some major concerns that should be addressed, including whether serum creatinine was measured at the same time as the 99mTc DTPA study.

Major Concerns:

1. I am curious why the authors chose the bedside Schwartz and EKFC equations as there are many other calculating equations that have been developed. In fact, many pediatric nephrologists have shifted to using the recently developed CKiD U25 creatinine and cystatin C based eGFR formulas (Pierce Kidney Int 99:948, 2021). It would be informative to compare the CKiD U25 creatinine based equation with the bedside Schwartz and EKFC equation. If the authors have cystatin C data, addition of a combined creatinine-cystatin C based calculation would also be informative.

Response:

We appreciate this important point regarding the choice of GFR estimating equations.

In Indonesia, the bedside Schwartz equation remains the standard equation recommended in National and Ministry of Health pediatric CKD guidance and is the routine formula used in our hospital for clinical decision-making in children. Thus, any validation in our local CAKUT population must, as a first priority, address how well this widely used equation reflects measured GFR (mGFR) in our patients. The EKFC equation was selected because it represents a contemporary “full age spectrum” creatinine-based formula, designed to harmonize GFR estimation from childhood into adulthood and increasingly recommended in European settings. It uses age (and not height) as a primary covariate, which is attractive for laboratory-based reporting and for regions where height measurement is not systematically captured in electronic records. Evaluating whether EKFC can be safely adopted in our CAKUT population has immediate practical implications for laboratories in Indonesia.

Although we agree that the CKiD U25 equations (creatinine-only and creatinine–cystatin C) show excellent performance in CKD cohorts, they have not yet been incorporated into Indonesian practice guidelines or routine laboratory reporting.

More importantly, serum cystatin C is not routinely available in our institution and in most Indonesian hospitals. It is substantially more expensive than creatinine and currently not covered by the national health insurance, which severely limits feasibility for routine clinical use in CAKUT patients. For the purpose of this retrospective study, we therefore did not have cystatin C measurements to compute combined creatinine–cystatin C equations.

Finally, we fully agree with the reviewer that validation of CKiD U25 creatinine-based in Indonesian children, including defining appropriate population-specific creatinine coefficients (analogous to population-specific “Q” values in EKFC), is an important and logical next step. We have add the CKiD U25 creatinine evaluation performance I this manuscript.

(page 3 line 1-2, page 4 line 21, page 5 line 3-4)

2. Given that this is a retrospective study, I am wondering if it is routine for all children with CAKUT to undergo mGFR measurements? Could there be selection bias in favor of discordant mGFR vs. eGFR in your cohort due to the inclusion of only patients who underwent 99mTc DTPA?

Response:

Thank you for raising this important issue of potential selection bias.

In our center, Tc-99m DTPA mGFR is recommended for all children with CAKUT when accurate GFR measurement is required for clinical decisions (e.g., pre-intervention assessment, significant structural abnormalities, or concern for CKD progression). During the study period, all newly diagnosed CAKUT patients were offered Tc-99m DTPA mGFR measurement as part of routine work-up.

Approximately 19% of children with CAKUT did not undergo Tc-99m DTPA, primarily due to: parental refusal of nuclear medicine procedures, logistic constraints, or scheduling issues rather than suspicion of discrepant eGFR. We agree that some degree of selection bias cannot be excluded. Those who accepted a nuclear medicine procedure may differ in unmeasured ways (e.g., disease severity, family preference, or socioeconomic factors) from those who declined.

To address this, we have now:

1. Explicitly acknowledged possible selection bias in the Limitations section and clarified that our findings strictly apply to CAKUT children for whom mGFR was clinically indicated and successfully obtained.

2. Emphasized that the clinical indication for Tc-99m DTPA was driven by the presence of CAKUT and need for accurate GFR, rather than by prior suspicion of a discrepancy between eGFR and mGFR. Therefore, while selection bias may exist, we believe it is unlikely to be systematically biased toward highly discordant eGFR/mGFR pairs.

We have added clarifying sentences on this point in the Methods and Limitations sections.

(page 4 line 5-10, page 12 line 4-13)

3. Was the serum creatinine used to calculate the bedside Schwartz and EKFC eGFR taken at the same time that the 99mTc DTPA was done?

Response:

Yes. We appreciate the chance to clarify this important methodological detail.

In our institutional protocol for nuclear renography, serum creatinine is drawn immediately before the Tc-99m DTPA injection, during the same visit and on the same day as the mGFR measurement.

For this study, only creatinine values obtained as part of the same visit as Tc-99m DTPA were used to calculate bedside Schwartz and EKFC eGFR. No creatinine measurements fro

---

## [Decision Letter · Decision Letter 1]

22 Dec 2025

Dear Dr. Aryati,

Thank you for submitting your manuscript to PLOS ONE. After careful consideration, we feel that it has merit but does not fully meet PLOS ONE’s publication criteria as it currently stands. Therefore, we invite you to submit a revised version of the manuscript that addresses the points raised by Reviewer 1 regarding the methods section, especially the inclusion and exclusion criteria.

We look forward to receiving your revised manuscript.

Kind regards,

Weining Lu, MD

Academic Editor

PLOS One

Journal Requirements:

Reviewers' comments:

Reviewer's Responses to Questions

**Comments to the Author**

Reviewer #1: (No Response)

2. Is the manuscript technically sound, and do the data support the conclusions?

Reviewer #1: Yes

3. Has the statistical analysis been performed appropriately and rigorously?

Reviewer #1: Yes

4. Have the authors made all data underlying the findings in their manuscript fully available?

Reviewer #1: Yes

5. Is the manuscript presented in an intelligible fashion and written in standard English?

Reviewer #1: Yes

Reviewer #1: This is a nice revision and the authors addressed my concerns. My only suggestion is to be more clear in the methods section regarding the inclusion criteria. I appreciate the additional demographic information in Table 1, but it would still be helpful if the authors could more explicitly spell out their inclusion criteria for this study, including how they defined “CAKUT”, what age ranges they considered for inclusion, and the exact time range that was considered. In addition, the authors say that 65 (19%) were excluded due to incomplete data – were these the 19% that did not undergo 99mTc DTPA? If so, I think it would be helpful to mention this in the methods section in addition to the nice commentary in the discussion.

**Do you want your identity to be public for this peer review?** For information about this choice, including consent withdrawal, please see our Privacy Policy

Reviewer #1: No

---

## [Author Response · Author response to Decision Letter 2]

22 Dec 2025

Dear Academic Editor and Reviewer,

We sincerely thank Reviewer 1 for the positive assessment of our revision and for the helpful suggestion to clarify the inclusion criteria in the Methods section. In response, we have expanded the “Population Study” subsection to explicitly describe (i) how CAKUT was defined and operationalized in our electronic medical record (EMR) search, (ii) the eligible age range, (iii) the exact study time window, and (iv) the relationship between the excluded cases and absence of paired 99mTc-DTPA mGFR measurements. We have also added a brief statement in the Methods reinforcing that the excluded 19% primarily reflects children without a completed 99mTc-DTPA assessment (and therefore without mGFR), complementing our discussion of this issue in the Limitations section.

Reviewer #1, Comment:

“My only suggestion is to be more clear in the methods section regarding the inclusion criteria... including how they defined ‘CAKUT’, what age ranges they considered for inclusion, and the exact time range that was considered. In addition, the authors say that 65 (19%) were excluded due to incomplete data – were these the 19% that did not undergo 99mTc DTPA? If so, I think it would be helpful to mention this in the methods section...”

Response:

We agree and appreciate this suggestion. We have revised the Methods to make the inclusion criteria explicit and reproducible. Specifically, we now state that we identified all pediatric patients with a *primary* diagnosis of CAKUT documented in the EMR within the study period and then included only those with a complete paired dataset for same-day serum creatinine and 99mTc-DTPA measured GFR (mGFR), alongside key demographic variables required for eGFR calculation.

• Definition of CAKUT: We now explicitly define CAKUT in the Methods as congenital structural anomalies of the kidneys and/or urinary tract documented in the EMR, and we clarify that CAKUT subtypes were abstracted from the clinical diagnosis and categorized as obstructive vs non-obstructive phenotypes (as summarized in Table 1).

• Age range: We now explicitly restate that eligible participants were children aged 6 months to 18 years at the time of 99mTc-DTPA renography.

• Time window: We now specify the exact time window used for EMR identification and data extraction (from January 1, 2017 through March 15, 2024; date of data access/extraction).

• Excluded 19% and 99mTc-DTPA: Yes—these 65 excluded cases were primarily those without a completed 99mTc-DTPA study and therefore lacking mGFR, and/or missing other required variables (e.g., height or same-day creatinine). We now state this explicitly in the Methods and briefly summarize common reasons (e.g., parental refusal of nuclear medicine procedures, logistic constraints, or scheduling issues), consistent with the Limitations section.

Text added/updated in the manuscript (Methods – “Population Study” subsection):

“This retrospective cohort study used electronic medical record (EMR) data from Dr. Soetomo Academic General Hospital to identify pediatric patients with a *primary* diagnosis of congenital anomalies of the kidney and urinary tract (CAKUT) from January 1, 2017 to March 15, 2024 (date of data extraction). CAKUT was operationalized as a congenital structural anomaly of the kidneys and/or urinary tract recorded as the primary clinical diagnosis in the EMR; subtypes were abstracted from the treating clinicians’ documented diagnosis and grouped as obstructive and non-obstructive CAKUT phenotypes (Table 1). Eligible participants were children aged 6 months to 18 years who underwent technetium-99m diethylenetriaminepentaacetic acid (99mTc-DTPA) renography for measured GFR (mGFR) and had serum creatinine assessed on the same day immediately preceding renography. We excluded records with missing data needed to compute eGFR or evaluate outcomes (age, sex, height, serum creatinine, mGFR, and urine albumin-to-creatinine ratio [ACR]). Of 341 CAKUT records screened, 65 (19%) were excluded because a complete paired dataset was not available, most commonly because 99mTc-DTPA was not performed, and/or because other key variables (e.g., height or same-day creatinine) were missing.”

We hope these revisions address the reviewer’s request for greater methodological clarity and improve reproducibility of the study.

Sincerely,

Aryati Aryati

(Corresponding Author)

---

## [Editor Report · Decision Letter 2]

26 Dec 2025

Performance of Glomerular Filtration Rate Equations Using Serum Creatinine in Children with Congenital Anomalies of the Kidney and Urinary Tract

PONE-D-25-32018R2

Dear Dr. Aryati,

We’re pleased to inform you that your manuscript has been judged scientifically suitable for publication and will be formally accepted for publication once it meets all outstanding technical requirements, including correction of any typos in your manuscript as described in the comment section below.

Kind regards,

Weining Lu, MD

Academic Editor

PLOS One

Additional Editor Comments:

Please review your final manuscript carefully and correct any typo (e.g., typo "CAUT" should be corrected as "CAKUT" on page 3, in the Material and Methods Population Study section).
---

## [Editor Report · Acceptance letter]

PONE-D-25-32018R2

PLOS One

Dear Dr. Aryati,

I'm pleased to inform you that your manuscript has been deemed suitable for publication in PLOS One. Congratulations! Your manuscript is now being handed over to our production team.

Kind regards,

on behalf of

Dr. Weining Lu

Academic Editor

PLOS One